# Neuroticism Predicts Subsequent Risk of Major Depression for Whites but Not Blacks

**DOI:** 10.3390/bs7040064

**Published:** 2017-09-21

**Authors:** Shervin Assari

**Affiliations:** 1Department of Psychiatry, Medical School, University of Michigan, Ann Arbor, MI 48109, USA; assari@umich.edu; Tel.: +1-734-232-0445; Fax: +1-734-615-8739; 2Center for Research on Ethnicity, Culture, and Health, School of Public Health, University of Michigan, 1415 Washington Heights, 2858 SPH1, Ann Arbor, MI 48109-2029, USA

**Keywords:** ethnic groups, African Americans, whites, neuroticism, depression

## Abstract

Cultural and ethnic differences in psychosocial and medical correlates of negative affect are well documented. This study aimed to compare blacks and whites for the predictive role of baseline neuroticism (N) on subsequent risk of major depressive episodes (MDD) 25 years later. Data came from the Americans’ Changing Lives (ACL) Study, 1986–2011. We used data on 1219 individuals (847 whites and 372 blacks) who had data on baseline N in 1986 and future MDD in 2011. The main predictor of interest was baseline N, measured using three items in 1986. The main outcome was 12 months MDD measured using the Composite International Diagnostic Interview (CIDI) at 2011. Covariates included baseline demographics (age and gender), socioeconomics (education and income), depressive symptoms [Center for Epidemiologic Studies Depression Scale (CES-D)], stress, health behaviors (smoking and driking), and physical health [chronic medical conditions, obesity, and self-rated health (SRH)] measured in 1986. Logistic regressions were used to test the predictive role of baseline N on subsequent risk of MDD 25 years later, net of covariates. The models were estimated in the pooled sample, as well as blacks and whites. In the pooled sample, baseline N predicted subsequent risk of MDD 25 years later (OR = 2.23, 95%CI = 1.14–4.34), net of covariates. We also found a marginally significant interaction between race and baseline N on subsequent risk of MDD (OR = 0.37, 95% CI = 0.12–1.12), suggesting a stronger effect for whites compared to blacks. In race-specific models, among whites (OR = 2.55; 95% CI = 1.22–5.32) but not blacks (OR = 0.90; 95% CI = 0.24–3.39), baseline N predicted subsequent risk of MDD. Black-white differences in socioeconomics and physical health could not explain the racial differences in the link between N and MDD. Blacks and whites differ in the salience of baseline N as a psychological determinant of MDD risk over a long period of time. This finding supports the cultural moderation hypothesis and is in line with other previously reported black–white differences in social, psychological, and medical correlates of negative affect and depression.

## 1. Background

Neuroticism (N) is a psychological trait with considerable public health significance [1]. N is a relatively stable personality trait that reflects individuals’ tendency to respond to threats with negative emotions. Individuals with high N show intense and frequent emotional reactions to minor challenges in their lives [2]. Trait N also increases sensitivity to negative emotional information [3,4]. Therefore, individuals with high N frequently experience negative emotional arousal, which will increase their risk for a wide range of negatively charged emotions such as sadness, anger, anxiety, fear, worry, frustration, and loneliness [5,6,7].

The literature has documented a robust link between N and several undesirable physical and mental health outcomes such as anxiety, heart disease, stroke, hypertension, multi-morbidity, and mortality [1,2,8,9,10]. N also predicts health behaviors, health service use, and quality of life [2,6]. High N is also a vulnerability factor for frequency and severity of depression [11,12,13,14,15].

A recent study, however, suggests that N may not be universally harmful [9,16,17,18,19]. While N predicts a higher risk factor for cardiovascular mortality in women with low socioeconomic status (SES), it may be protective for women with high SES [9]. Park et al. found that N alters the link between social support and health in Japanese but not American individuals [20]. To explain the above findings, Kitayama has argued that N may be protective in some contexts, as it may reflect the sensitivity of people to potential costs associated with environmental exposures [20]. According to this argument, in some but not all contexts, high N means avoidance of exposure, which has positive health effects [21].

Although still new, a few studies have suggested that the health effects of N [20] and other negative affective domains such as depression [22,23], anger [24,25], and hostility [24] depend on race, ethnicity, and culture. This is in line with the “differential effects hypothesis” [26], which suggests that social, psychosocial, and behavioral mechanisms that ultimately shape the health and illness of populations are not universal but population-specific. This hypothesis conceptualizes race as an effect modifier that alters the effect of the same risk factors on an outcome [27]. Considerable empirical support exists for this hypothesis, suggesting that group differences in the health effects of psychosocial constructs such as depression and negative affect are rules rather than exceptions [28,29].

While some evidence has suggested that context alters the health correlates of N [9,20] and some research has shown that baseline N may have implications for subsequent mental health decades later [30], it is still unknown whether blacks and whites differ in the predictive role of high N on subsequent risk of depression years later. The aim of the current study was to compare blacks and whites for the predictive value of baseline N for subsequent risk of MDD 25 years later. To generate nationally representative results, we used data from a national sample of adults in the USA.

## 2. Methods

### 2.1. Design and Setting

Data came from the Americans’ Changing Lives (ACL) study, 1986–2011. ACL is a longitudinal cohort study with a nationally representative sample. The detailed methodology of the ACL study is documented elsewhere in the literature [31,32].

### 2.2. Ethical Considerations

The University of Michigan Institutional Review Board approved the study protocol (HUM00051735). The study protocol was in accordance with The Code of Ethics of the World Medical Association (Declaration of Helsinki, Edinburgh 2000 revision). All participants provided informed written consent.

### 2.3. Sampling and Participants

The ACL cohort study has followed a nationally representative sample of American adults 25 years old and older from 1986 to 2011. The ACL has applied a stratified multistage probability sampling of the U.S. population. The baseline population of ACL in 1986 included 3617 adults 25 years old and older who were non-institutionalized and were sampled from the continental USA. The study has oversampled blacks and older adults (age > 60) at twice the rate of others.

### 2.4. Analytical Sample

The current study used whites and blacks who were in the study in wave 1 as well as wave 5 of the study (1986 and 2011). From the 3361 individuals (2205 whites and 1156 blacks) who were followed from 1986 for up to 25 years, 1219 individuals (847 whites and 372 blacks) had data on MDD in 2011 and entered our analysis.

### 2.5. Data Collection

Baseline data on demographics (age, gender), socioeconomics, depressive symptoms, and health status were collected via face-to-face interviews in wave 1 in 1986. Baseline demographic factors (age and gender), socioeconomic status (education and income), depressive symptoms [Center for Epidemiologic Studies-Depression scale (CES-D)], stress, health behaviors (smoking and drinking), and physical health (Chronic Medical Conditions [CMC], obesity, and Self-Rated Health [SRH]) were potential confounders. The main predictor of interest was baseline N. The main outcome was clinical MDD 25 years later, measured by Composite International Diagnostic Interview (CIDI) in 2011.

### 2.6. Measures

*Race and Ethnicity.* The ACL study used multiple survey items in defining the race and ethnicity of the participants at baseline. First, participants were required to answer an open-ended question: “In addition to being American, what do you think of as your ethnic background or origins?” Then, a multiple-choice question was asked with the possibility that multiple categories could be chosen: “Are you white, black, American Indian, Asian, or another race?” Those who answered with more than one non-white group were required to say which one “best described” their race. The participants were also asked about their father’s last name and the state or country in which the participant, participant’s mother, and participant’s father were born. Finally, the participants were required to answer whether they were of “Spanish or Hispanic descent, that is, Mexican, Mexican American, Chicano, Puerto Rican, Cuban, or Other Spanish?” Race categories were constructed based on the participants’ responses to the above questions: “Non-Hispanic White”, “Non-Hispanic Black”, “Non-Hispanic Native American”, “Non-Hispanic Asian”, and “Hispanic”. We only included Non-Hispanic White and Non-Hispanic Black individuals in our analysis [22,23].

#### 2.6.1. Demographic Factors

The ACL collected data on age (in years) and gender (considering male as the reference) at the first wave in 1986.

#### 2.6.2. Socio-Demographics (SES)

The ACL collected baseline data on education (>12 years of schooling) and household income (a continuous variable).

#### 2.6.3. Health Behaviors

Data were collected on smoking (i.e., tobacco use) and drinking (i.e., alcohol consumption) in 1986. Information was collected on self-reported history of smoking, using a dichotomous variable (current smoker = 1, never or ex-smoker = 0). Self-reported alcohol use was measured by asking whether or not the respondent currently drinks (1 = current drinker; 0 = non-drinker) [33]. Two dummy variables were created for smoking and drinking.

#### 2.6.4. Stressful Life Events (SLE)

Data were collected data on the number of negative life events in the past three years. We used a measure that accords well with the current standards of measurement for major/traumatic events [34].

#### 2.6.5. Neuroticism (N)

Neuroticism was measured with three indicators that asked respondents to indicate the extent to which they (1) consider themselves moody; (2) felt fed up; or (3) were tense or high-strung. The exact items read (1) Does your mood often go up and down? (2) Do you often feel fed up? (3) Would you call yourself tense or “high-strung”? Response categories for all items were “no,” (1) “sometimes;” (3) and “yes;” (5) with a higher score on these indicators reflecting higher levels of neuroticism.

#### 2.6.6. Depressive Symptoms

The 11-item CES-D scale was used to measure depressive symptomatology: Participants were asked about depressed mood, feelings of guilt and worthlessness, feelings of helplessness and hopelessness, as well as restless sleep [10,15]. Each item was scored from 1 to 3. We calculated a mean score ranging from 1 to 3. Then we used a Z score that has a mean of 0 and a standard deviation of 1. CES-D has high validity and reliability [35,36,37,38].

#### 2.6.7. Obesity

Obesity was based on self-reported weights and heights. Weight and height were originally collected in pounds (1 pound = 0.453 kg) and feet (1 foot = 0.3048 m)/inches (1 inch = 0.0254 m). Obesity was defined as a body mass index (BMI) of equal or larger than 30 kg/m^2^. BMI based on self-reported data closely correlates with BMI based on direct height and weight measures [39].

#### 2.6.8. Chronic Medical Conditions (CMC)

Number of CMC was collected using self-reported, physician-diagnosed medical conditions. Participants were asked if they have been informed by a healthcare provider about having one or more of seven focal conditions. The conditions listed were hypertension, diabetes, heart disease, stroke, cancer, chronic lung disease, and arthritis [32,40].

#### 2.6.9. Major Depressive Disorder (MDD)

The main outcome was endorsement of DSM IV criteria for clinical diagnosis of MDD, measured using the Composite International Diagnostic Interview (CIDI). The CIDI—originally designed for the World Mental Health project—measures endorsement of criteria diagnosis of recent and lifetime non-psychotic mental disorders. The CIDI is administered by trained lay interviewers. CIDI and Structured Clinical Interviews for DSM-IV diagnoses (SCID) have shown good concordance for diagnosis of major depressive disorders, as reflected by the area under the receiver operating characteristic curve (AUC) [41,42,43,44].

### 2.7. Statistical Analysis

Stata 13.0 (Stata Corp., College Station, TX, USA) was used for data analysis. Stata enabled us to account for the complex sample design of the ACL study by considering sampling and non-response weights. Taylor series linearization was used to estimate standard errors (SE). Subpopulation survey commands were used for data analysis. The *p*-value significance cut-off point was considered 0.05 and 0.1 for main effect and interactions, respectively.

Descriptive and frequency tables were used for univariate analysis. Four survey logistic regression models were used for multivariable analysis. In all logistic regression models, baseline N was the main independent variable, subsequent MDD in 2011 was the main outcome, and demographics, SES, depressive symptoms, stress, health behaviors, and physical health were covariates. First we ran two models in the pooled sample. In *Model 1*, we only tested the main effect of N and covariates. In *Model 2*, we also included the N by race interaction. Then we conducted the same models specific to whites (*Model 3*) and blacks (*Model 4*). Odds Ratio and standard errors, 95% CI, and p values were reported for each variable.

## 3. Results

### 3.1. Descriptive Statistics

Table 1 presents descriptive statistics in the pooled sample, and based on race. Independent samples’ *t*-test and chi-squared test showed lower education, income, and drinking but higher N, SLE, depressive symptoms, CMC, obesity, and smoking among blacks compared to whites.

### 3.2. Models in the Pooled Sample

Table 2 summarizes the results of two logistic regression models on the association between baseline N and subsequent risk of CIDI-based MDD 25 years later in the pooled sample, net of baseline controls. Based on *Model 1*, in the pooled sample, higher N at baseline (OR = 2.23, 95% CI = 1.14–4.34) was positively associated with risk of CIDI-based MDD 25 years later in 2011. Based on *Model 2*, in the pooled sample, N at baseline symptoms showed a marginally significant interaction with race (OR = 0.37, 95% CI = 0.12–1.12), suggesting a stronger effect of baseline N on subsequent MDD 25 years later for whites compared to blacks. Such interaction was the net of socioeconomic and health in 1986 (Table 2).

### 3.3. Models among Whites and Blacks

According to *Model 3*, higher level of N at baseline was a significant predictor of MDD 25 years later in 2011 in whites (OR = 2.55, 95% CI = 1.22–5.32). According to *Model 4*, higher N (OR = 0.90, 95% CI = 0.24–3.39) was not predictive of MDD 25 years later in 2011 for blacks (Table 3).

## 4. Discussion

We found that high N at baseline better predicted subsequent risk of clinical MDD 25 years later in whites compared to blacks. In race-specific models, baseline N predicted MDD for whites but not blacks.

Previous research has suggested that negative affect in general [22,23,25] and N in particular [9,20] may have group-specific rather than universal health effects. Assari has shown that depressive symptoms predict all-cause [22] and cause-specific [23] mortality in whites but not blacks. Race has been shown to interact with baseline depressive symptoms on risk of all cause and cause-specific mortality, suggesting a stronger effect of baseline depressive symptoms on mortality for whites compared to blacks [22,23]. Hostility and anger have also predicted cardiovascular mortality in whites but not blacks [24]. Park et al. found that N alters the link between social support and health in Japanese but not American individuals [20], and anger may even be linked to better health in some cultures [25]. Park et al. showed that for white Americans, lower social standing was associated with greater expression of anger, while for Japanese individuals high social status was associated with more anger expression. While for white Americans, anger expression was predicted by subjective social status, for Japanese individuals the objective social status predicted anger expression [45].

To explain the mixed findings on the health effects of N, Kitayama et al. have argued that N may be protective in some contexts, as it may reflect the sensitivity of people to potential costs associated with environmental exposures [20]. Thus, at least in some contexts, individuals with high N may be able to avoid exposure, which may have positive health effects [21]. Jackson has attributed the differential effects of stress and chronic disease on depression of blacks and whites to differential behavioral coping mechanisms that populations use [46,47,48,49]. Assari has, however, shown that the moderating effects of contextual factors such as race, ethnicity, and culture hold for a wide range of psychosocial domains and health outcomes [22,23,28,29,50,51,52,53,54,55,56].

Some of these findings may be, at least in part, due to the effects of culture in shaping emotions [57,58,59]. Based on the behavioral adjustment model, Kitayama et al. proposed that, if combined with one’s flexibility to change behavior in face of need, N can alert individuals to potential threats in the environment and, as a consequence, can represent lower levels of exposure to interpersonal and environmental threats [60,61]. It is, however, still unknown by which exact mechanism race, ethnicity, gender, and culture alter the health outcomes associated with psychosocial factors such as N and depression.

A wide range of subjective measures of mental health have shown stronger links with objective measures of physical and mental health in whites than blacks [50,56,62,63]. Race may also alter how traits that reflect personality or distress covary with particular psychiatric disorders such as MDD based on structural interviews [62]. Perceived mental health better reflects psychiatric disorders among whites than in minorities such as blacks [56]. Concordance between CES-D score and clinical depression also depends on contextual factors such as ethnicity, culture, and social class [10,64]. Similar findings have been reported on the SLE–MDD link [65,66].

In the pooled sample, we could find a significant interaction between race and baseline N on subsequent MDD risk, suggesting that the predictive role of baseline N for MDD is larger for whites compared to blacks. Such black-white difference was not due to racial differences in socioeconomic and health status. Our race-specific models also showed that baseline N is predictive of subsequent risk of MDD among whites but not blacks, net of SES and health status at baseline. These findings are in line with previous findings on the differential role of CES-D on subsequent risk of MDD [10] as well as the black-white differences in social, psychological, and medical correlates of negative affect, MDD, and depressive symptoms [40,52,53,54,55,65,66,67].

Based on our findings, compared to whites, blacks had higher N in 1986 and lower risk of endorsement of CIDI-based MDD 25 years later in 2011. The literature has documented systematically weaker effects of a wide range of psychosocial factors on health outcomes for blacks compared to whites [50,51,52,53,54]. We cannot rule out that at least some of the findings may be due to measurement issues regarding diagnosis of clinical depression among blacks [68,69]. There is some previous research suggesting that CIDI validity may differ for blacks and whites [70,71,72], which may in turn result in differential correlates of depression based on race. Finally, it is not clear whether racial groups are similar in terms of the stability of personality traits such as N over time [72].

There are a number of limitations to our study. First, we used a three-item measure for N, which may have differential validity across racial groups. As a result, there is a need for replication of these findings using standardized measures of N. Secondly, in our analysis we conceptualized N, SES, stress, depressive symptoms, and physical health as fixed factors; however, all these constructs are subject to change over a long period of time. Third, the sample size was not balanced between whites and blacks, and the low sample size of blacks may have caused statistically under-powered study for blacks. Fourth, we did not control for diagnosis of MDD, mental health care use, or anti-depressant prescription in this study. Fifth, the 25-year follow-up period may have caused selective attrition in this study. Due to the higher mortality of blacks, race was not independent of attrition in this study. Future research may consider replication of the current findings using shorter follow-up periods. Despite these limitations, this is one of the first studies on black-white differences in the predictive role of baseline N on subsequent risk of MDD diagnosis based on CIDI 25 years later in a nationally representative sample of American adults.

In conclusion, baseline N predicts subsequent risk of MDD in whites but not blacks. Racial differences in the effect of baseline N on subsequent risk of CIDI-based MDD are in line with previous findings on the cultural moderation of correlates of negative affect (N, hostility, anger, depression, depressive symptoms), as well as racial differences in the complex links between socioeconomic status, negative affect, and physical health [73]. Future neuroscience research is needed to compare biological and social correlates of high N across racial groups.

## Figures and Tables

**Table 1 behavsci-07-00064-t001:** Descriptive statistics for baseline characteristics based on race.

	All (*n* = 1219)		Whites (*n* = 847)		Blacks (*n* = 372)	
	*n* (SE)	95% CI	*n* (SE)	95% CI	*n* (SE)	95% CI
Race *						
White	90.06 (0.01)	87.86–91.90	–	–	–	–
Black	9.94 (0.01)	8.10–12.14	–	–	–	–
Gender *						
Men	46.11 (0.02)	42.81–49.45	46.57 (0.02)	42.98–50.19	41.97 (0.03)	35.78–48.43
Women	53.89 (0.02)	50.55–57.19	53.43 (0.02)	49.81–57.02	58.03 (0.03)	51.57–64.22
Obese *						
No	86.78 (0.01)	84.70–88.61	87.23 (0.01)	84.82–89.30	82.69 (0.02)	77.64–86.79
Yes	13.22 (0.01)	11.39–15.30	12.77 (0.01)	10.70–15.18	17.31 (0.02)	13.21–22.36
Any chronic medical conditions *						
No	68.86 (0.02)	65.25–72.24	69.92 (0.02)	65.94–73.62	59.22 (0.03)	52.30–65.79
Yes	31.14 (0.02)	27.7634.75–	30.08 (0.02)	26.38–34.06	40.78 (0.03)	34.21–47.70
Drinking *						
No	32.66 (0.02)	28.72–36.86	31.31 (0.02)	27.19–35.76	44.84 (0.04)	37.66–52.24
Yes	67.34 (0.02)	63.14–71.28	68.69 (0.02)	64.24–72.81	55.16 (0.04)	47.76–62.34
Smoking *						
No	70.65 (0.02)	66.93–74.11	71.43 (0.02)	67.36–75.18	63.55 (0.03)	56.37–70.17
Yes	29.35 (0.02)	25.89–33.07	28.57 (0.02)	24.82–32.64	36.45 (0.03)	29.83–43.63
MDD *						
No	91.23 (0.01)	89.13–92.96	91.51 (0.01)	89.27–93.31	88.76 (0.02)	83.95–92.26
Yes	8.77 (0.01)	7.04–10.87	8.49 (0.01)	6.69–10.73	11.24 (0.02)	7.74–16.05
Neuroticism *						
Low	83.52 (0.01)	80.43–86.20	83.98 (0.02)	80.41–87.01	79.29 (0.02)	74.64–83.27
High	16.48 (0.01)	13.80–19.57	16.02 (0.02)	12.99–19.59	20.71 (0.02)	16.73–25.36
	% (SE)	95% CI	% (SE)	95% CI	% (SE)	95% CI
Age	47.77 (0.53)	46.69–48.84	47.96 (0.60)	46.75–49.17	46.33 (0.72)	44.89–47.78
Education (>12 years) *	12.53 (0.10)	12.34–12.73	12.69 (0.11)	12.48–12.90	11.37 (0.23)	10.90–11.84
Household income *	5.41 (0.09)	5.22–5.60	5.57 (0.10)	5.36–5.77	4.25 (0.18)	3.88–4.62
Stress *	0.88 (0.02)	0.84–0.92	0.88 (0.02)	0.84–0.92	0.87 (0.03)	0.81–0.94
Depressive symptoms *	−0.03 (0.02)	−0.08–0.02	−0.07 (0.03)	−0.13–0.02	0.28 (0.05)	0.18–0.38
Neuroticism *	−0.01 (0.03)	−0.06–0.04	−0.01 (0.03)	−0.07–0.05	0.01 (0.04)	−0.07–0.09

Notes: MDD: Major Depressive Disorder. * *p* < 0.05.

**Table 2 behavsci-07-00064-t002:** Summary of logistic regression models on the association baseline neuroticism and CIDI-based major depressive episode 25 years later based on race (*n* = 1219).

	OR (SE)	95% CI	*p*	OR		*p*
	Model 1			Model 2		
Race(Blacks)	0.88 (0.22)	0.54–1.45	0.612	1.23 (0.34)	0.70–2.16	0.455
Age	0.99 (0.01)	0.96–1.02	0.342	0.99 (0.01)	0.96–1.02	0.325
Gender (Women)	1.02 (0.25)	0.63–1.66	0.927	1.03 (0.25)	0.63–1.69	0.894
Education (>12 years)	0.99 (0.08)	0.85–1.17	0.941	1.00 (0.08)	0.85–1.17	0.964
Household income	0.98 (0.06)	0.86–1.12	0.778	0.98 (0.06)	0.86–1.11	0.751
Any chronic medical condition	1.18 (0.31)	0.70–1.99	0.530	1.20 (0.31)	0.71–2.03	0.477
Obese	1.42 (0.39)	0.81–2.48	0.215	1.41 (0.39)	0.81–2.45	0.219
Drinking	0.73 (0.16)	0.47–1.12	0.148	0.73 (0.16)	0.48–1.13	0.156
Smoking	1.55 (0.44)	0.87–2.75	0.130	1.53 (0.44)	0.86–2.73	0.141
Stress	1.36 (0.21)	0.99–1.86	0.054	1.36 (0.21)	1.00–1.85	0.050
Depressive symptoms	1.55 (0.18)	1.22–1.97	0.001	1.56 (0.18)	1.23–1.98	0.000
Neuroticism	2.23 (0.74)	1.14–4.34	0.020	2.51 (0.88)	1.24–5.08	0.012
Neuroticism × Race	-	-	-	0.37 (0.20)	0.12–1.12	0.077
Constant	0.10 (0.13)	0.01–1.49	0.092	0.09 (0.12)	0.01–1.41	0.085

Notes: CIDI: Composite International Diagnostic Interview.

**Table 3 behavsci-07-00064-t003:** Summary of logistic regression models on the association baseline neuroticism and CIDI-based major depressive episode 25 years later based on race (*n* = 1219).

	OR (SE)	95% CI	*p*	OR (SE)	95% CI	*p*
	Model 3			Model 4		
	Whites (*n* = 847)			Blacks (*n* = 372)		
Age	0.98 (0.02)	0.95–1.02	0.343	0.98 (0.03)	0.93–1.03	0.374
Gender (Women)	1.05 (0.30)	0.59–1.85	0.863	0.95 (0.59)	0.27–3.34	0.932
Education (>12 years)	1.01 (0.10)	0.83–1.22	0.929	0.91 (0.12)	0.71–1.18	0.467
Household income	0.97 (0.07)	0.84–1.13	0.685	1.06 (0.09)	0.89–1.26	0.524
Any chronic medical condition	1.31 (0.40)	0.71–2.42	0.376	0.63 (0.31)	0.23–1.68	0.347
Obese	1.33 (0.46)	0.66–2.69	0.421	2.07 (0.94)	0.82–5.20	0.118
Drinking	0.65 (0.15)	0.40–1.05	0.075	1.52 (0.83)	0.50–4.61	0.450
Smoking	1.42 (0.44)	0.76–2.63	0.267	2.13 (0.72)	1.08–4.23	0.031
Stress	1.38 (0.25)	0.96–2.00	0.084	1.32 (0.20)	0.97–1.79	0.073
Depressive symptoms	1.53 (0.20)	1.18–1.99	0.002	1.83 (0.35)	1.25–2.70	0.003
Neuroticism	2.55 (0.93)	1.22–5.32	0.014	0.90 (0.59)	0.24–3.39	0.871
Constant	0.09 (0.13)	0.00–1.87	0.116	0.24 (0.64)	0.00–54.25	0.596

Notes: CIDI; Composite International Diagnostic Interview.

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
