# Peer review of "Neuroticism Predicts Subsequent Risk of Major Depression for Whites but Not Blacks"

_behavsci, 2017, doi:10.3390/bs7040064_

Round 1

Reviewer 1 Report

This is an interesting article that examined baseline neuroticism as a predictor of depression risk in Blacks vs. Whites in a longitudinal study 25 years later. This study is important because it  is one of the first studies on Black-White differences in the predictive role of baseline neuroticism  on subsequent risk of major depression. There are some interesting findings such as the higher neuroticism score in Blacks. The study is carefully done as noted by the care to identify race.  The literature review is relevant and strong. There are a few issues that distract from an otherwise very strong study. First there is no mention of the subjects lost to followup and in particular Black subjects, which has been problematic in other such studies.  The authors have a number of health measures but make little attempt to relate them to depression risk  or how they may impact their findings. For example they note smoking and drinking findings but make no attempt to relate them to the presence of depression. Such analyses help to add to the public health significance such that the findings are not simply a methodological study. Finally the authors often leave articles out of their sentence and their absence make some of the sentences difficult to interpret. 

Author Response

We have mentioned "lost to followup" and differential attrition based on race as a limitation. Yes, attrition is higher in Blacks, and we mentioned this now.

In previous studies we have shown that the health factors may also differentially be associated with depression of Whites and Blacks.

We have added a little bit to our results on the effects of health on future depression.

Yes, the author is not a native English speaker. We sent the paper for language edit.

Reviewer 2 Report

The take-home message of this study is clearly expressed in its title Neuroticism Predicts Future Depression among Whites but Not Blacks. This longitudinal study compare Blacks and whites over 25 year period to determine the predictive role of baseline neuroticism (N) on subsequent risk for episodes of major depression. I thought there were some portions of the text that could better conform to standard English usage and I felt that the results section was somewhat sketchy in its presentation of statistical details and could have benefited by the inclusion of specific P values. However, these concerns are relatively minor. The main issue I have with the study, particularly the way that it is presented, has to do with the variable around which its core premise is constructed, neuroticism. Throughout the manuscript it is taken as a given that all the participants contributing to the data have been initially assessed with a valid measure of neuroticism. In the discussion section a number of possible limitations to the study are listed but the validity of the baseline neuroticism data is not included as a possible limitation. Neuroticism in this study was measured with three items using a Likert scale. The items were: 1. Is your mood often go up and down? 2. Do you often feel fed up? 3. Would you call yourself tense or "high-strung"? These items were taken directly from Hans Eysenck's EPQ-R. The EPQ-R is 100 item personality inventory that measures the three factors of extraversion, psychoticism and neuroticism. There are a total of 24 items that measure neuroticism and even in the 50 item short version of the EPQ there are 12 items to measure neuroticism. Neuroticism is considered to be a measure of emotional lability or sensitivity but there are a variety of aspects included in this construct. These include, anxiety, anger, depression, self-consciousness, moodiness and lack of self-confidence. No doubt, the three items that were used in this study do give some indication of a penchant for emotional lability. It is for that reason that I support the publication of the study. Unfortunately, the limitations of that initial survey do not allow probing which factors (i.e. depression, anxiety, self-confidence etc.) are more strongly associated with risk of major depression. I believe at minimum, the author needs to address this issue where he talks about other limitations to the study.

Author Response

We improved the language of the paper.

We also added to the results section. Usually CI replace with the p values. So, it is not accepted to report both CI and p in the text.

We added the measurment of the neuroticism variable using 3 items as a major limitation of this study. It is a short measure and it is not a comprehensive standard measure of neuroticism.